# Improvement on Flux Weakening Control Strategy for Electric Vehicle Applications

**Claudio Bianchini** *, **Giovanni Franceschini and Ambra Torreggiani**

Department of Engineering (DIEF), University of Modena and Reggio Emilia, 41121 Modena, Italy;
giovanni.franceschini@unimore.it (G.F.); ambra.torreggiani@unimore.it (A.T.)
*   Correspondence: claudio.bianchini@unimore.it

**Abstract:** This paper proposes an optimized flux weakening (FW) control strategy for interior permanent-magnet synchronous electric motor to address the critical issues that could occur under torque setpoint transition in flux weakening region, due, for example, to an emergency braking. This situation is typical in electric vehicles where the electrical machines operate over a wide speed range to reach high power density and avoid gearboxes. Two modified traditional flux weakening strategies are proposed in this paper to improve torque control quality during high speed torque transition. The proposed modified control strategies were validated both by Matlab/Simulink simulations, modeling the power train of a light vehicle application, and extensive experimental tests on a dedicated test bench.

**Keywords:** interior permanent magnet synchronous electrical machine; electric vehicle; flux-weakening; torque control loop





## 1. Introduction

In electric vehicle applications, interior permanent magnet synchronous machine is the most common motor topology for electric power train thanks to its inherently high torque density and high flux weakening capability [1]. The flux weakening capability is a key issue when the vehicle power train layout does not include a gearbox. In addition, the permanent-magnets increase both motor power factor and efficiency avoiding the magnetizing current. An adequate and optimized control strategy is of a paramount importance to achieve a wide speed range. From a general point of view it is possible to identify two operating regions for the power train: the constant torque region where the maximum torque per ampere (MTPA) strategy is adopted to minimize the Joule losses and the constant power region, where the motor operates under flux weakening (FW) control strategy to comply with voltage limit. Figure 1 shows the constant torque and constant power operating regions as a function of the mechanical angular speed of the interior permanent-magnet synchronous motor evaluated in this paper.

Flux weakening is obtained feeding the machine with an additional negative d-axis current to reduce the permanent magnet (PM) flux linkage. Thanks to this additional d-axis current, the total current vector is moved toward the $i_d$ axis as shown in Figure 2.

When the characteristic current $i_{ch}$, that is the center of the voltage ellipse, is outside the current limit circle, the maximum theoretical speed is finite, and its value decreases as the absolute value of the characteristic current increases. On the contrary when $i_{ch}$ is on or inside the current limit circle, the maximum theoretical speed is infinite. It has to be noted that, when it is inside the current limit circle, to obtain an infinite speed Maximum Torque per Volt (MTPV) control strategy must be implemented, Figure 2. This results in a lower torque, since the amplitude of the current vector reduces and, meanwhile, the control gets more complicated. It is apparent that, if possible, the machine design should comply with Battery Pack voltage and Inverter current limits. There are many studies regarding the

optimal permanent magnet quantity needed to obtain an optimal flux weakening behavior, the main results of these studies can be synthesized in a relation between the inductance on the d-axis and a the permanent magnet flux linkage that place the characteristic current close to the edge of the current limit circle in the $i_{sd} - i_{sq}$ plane.

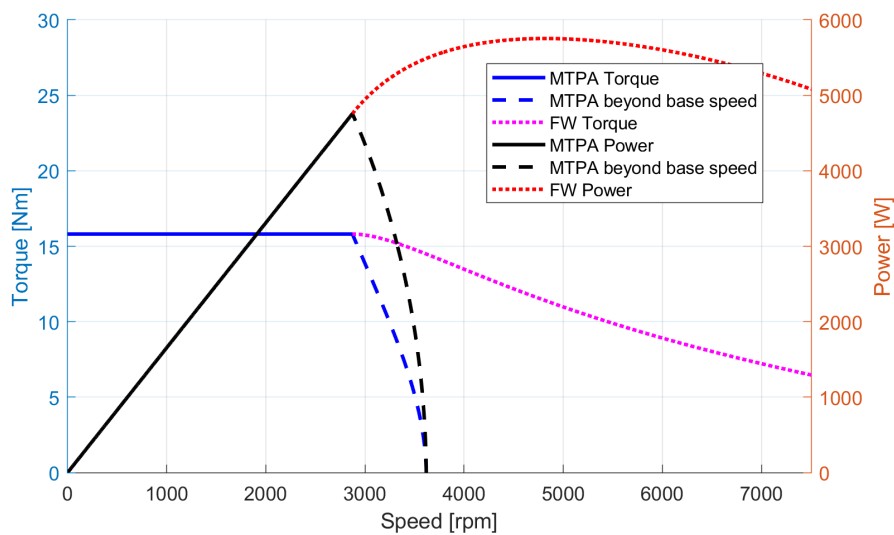

**Figure 1.** Constant torque and power regions for the reference motor under test.

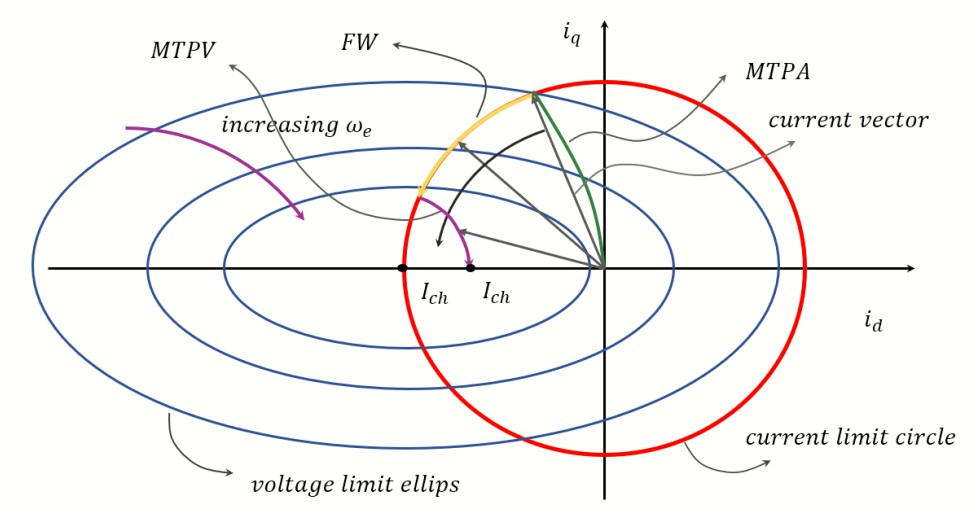

**Figure 2.** $i_d - i_q$ plane, MTPA, FW and MTPV control strategies.

This design approach allows the maximum exploitation of the electric power system and avoids the MTPV region [2–4] Thus, machines which avoid MTPV operation could be also considered for electric vehicle applications

Other control design approaches that implement all the three different control strategies were developed by different authors [5], this control scheme could be useful when the characteristic current of the electric machine falls inside the current limit circle.

In recent years, other different approaches were developed based on low coercive force of permanent-magnets [6], but their application is still uncommon. Some authors propose also mechanical solutions to reduce the permanent-magnets flux at high speed creating a different mechanical gap between rotor and stator in the axial direction as the speed increases [7], in this way it is possible to obtain flux weakening without any control strategy implemented on the drive.

Other works try to develop flux weakening algorithms to avoid the need of any magnetic machine model [8]. There are also some examples of control schemes based on different state variables (linkage flux instead of stator currents [9], or other works focused on loss minimization strategy [10].

In any case, when a wide-speed range is required and rare-earth PMs are employed, the current vector at maximum speed is very close to the $i_{sd}$ axis.

In the case of electric power train there are different current limit circles for the different duty types. In the case of continuous duty type (S1), the current limit circle is typically limited by the thermal behaviour of the electrical machine, while in short time over-current, or over-torque, (duty type—S2) the current limit circle is typically given by the inverter current limit, therefore a trade-off for the characteristic current is necessary.

In this study, the characteristic current of the electric motor is chosen in such way that it falls in the middle of the two current limits corresponding to the continuous duty type (S1) and the short-time duty (S2). The choice of the PMs volume must take into account not only the characteristic current, but also the demagnetization issues and the requirements linked to the torque density and the efficiency of the system. These latter considerations imply a design with a high PM flux that requires high negative d-axis current component at high speed to counteract the PMs back—electromotive force (Back-EMF). This corresponds to a current vector with an angle close to 180° and a high amplitude, even if a low torque setpoint is required.

Several FW control strategies [11], have been proposed recently; nevertheless, even if they are extensively tested in motor operation, few studies [12,13] and papers investigate on the effects of abrupt torque setpoint transition from motor operation to braking one in deep flux weakening region [14] as a consequence of a suddenly let go of the throttle.

In the case of electric vehicle, the control strategy is usually carried out with a torque feedback control, while the driver adjusts the vehicle speed with the throttle. When a substantial torque transition is required (e.g., during braking), the torque setpoint moves from its maximum value to zero and, if the motor is operating in the FW region, the d-axis current component must be properly controlled to counteract the flux linkage of PMs.

This working condition is critical, since the current vector must be kept ideally very close to the d-axis position 1, see Figure 3. In this operating point, even a small error in the electrical angle (e.g., due to the position sensor on the motor or to the delay introduced in the PWM period) could move the current vector from the motor operating region, position 1 of Figure 3, to the generator operation region, position 2 of Figure 3, producing unwanted additional braking torque or vibrations.

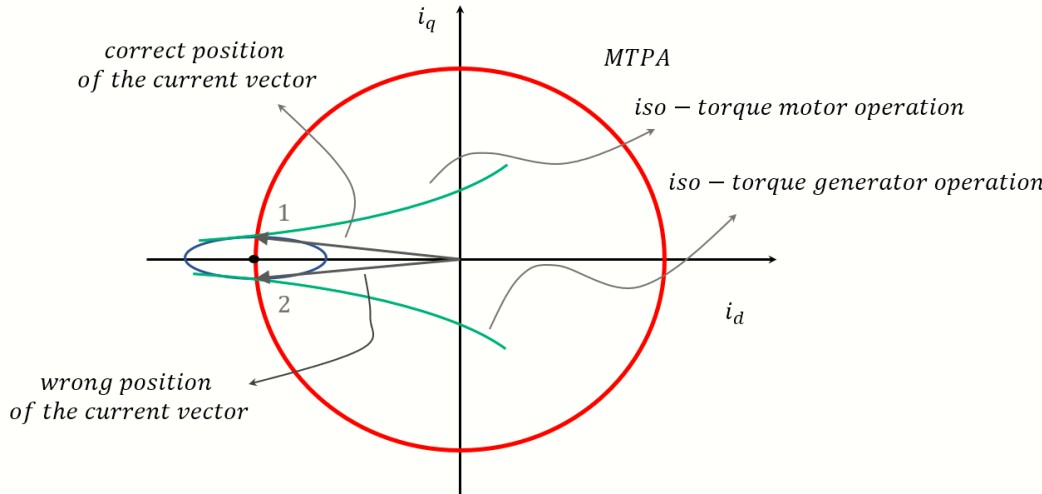

**Figure 3.** Current vector position at high-speed operations where: 1. motor operating region, 2. generator operating region.

Moreover, during fast torque setpoint transitions, the error of the current angle due to iron losses changes its sign [15], introducing additional issues in the managing of the energy-flow from the motor to the battery pack. The latter issue has to be addressed to avoid hazardous operation or overcharge of the battery pack.

In [16] author proposed a FW control scheme based on the detection of the $i_{sd}$ current error. The $i_{sd}$ axis error is used to decrease the q-axis voltage setpoint. Figure 4 shows essentially a modification of the classical feed forward (FF) vector control method. This control scheme allows a smooth and fast transition from the MTPA strategy to FW control strategies, without the need of an accurate knowledge of the motor parameters or look-up tables (LUT) of the machine magnetic model.

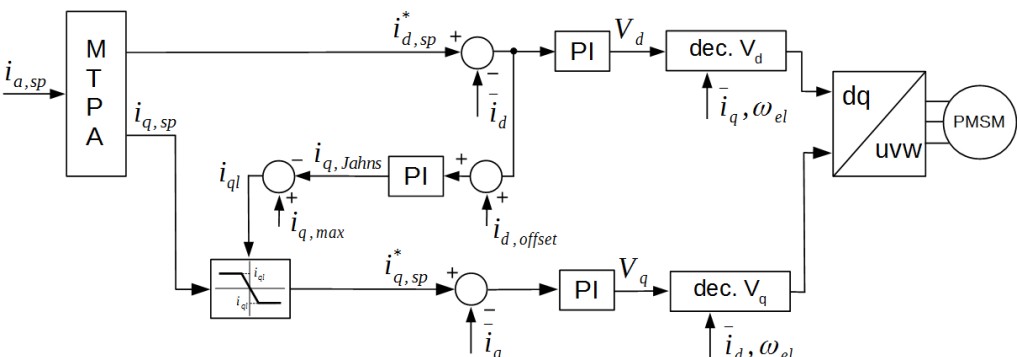

**Figure 4.** Traditional control scheme proposed in [16] based on $i_{sd}$ current error.

The FW control scheme of [17] calculates a suitable $i_{sd}$ setpoint by means of the outer voltage loop. As a consequence the voltage vector is forced within the limits represented by the Voltage DC BUS. This control scheme of Figure 5 allows the maximum voltage exploitation, nevertheless in this approach no specific strategy is deeply investigated to manage a possible fast torque transition at high speed during FW operation.

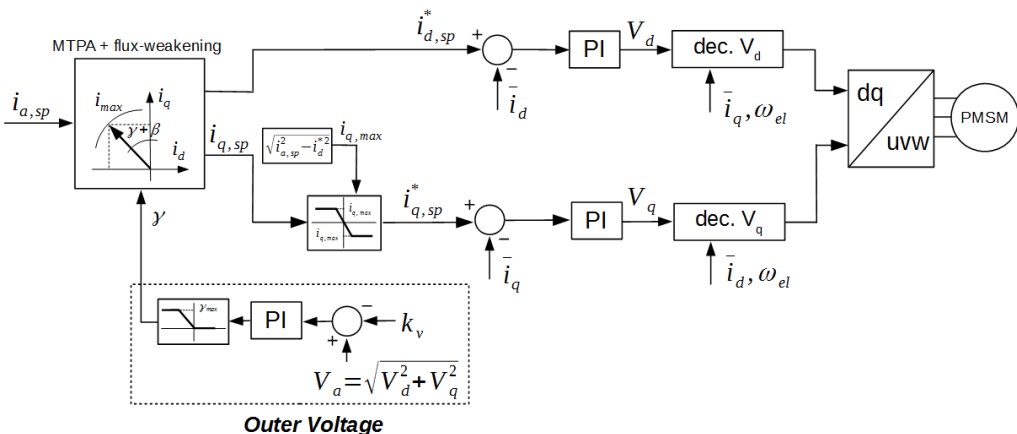

**Figure 5.** Traditional FW control scheme proposed in [17] based on outer voltage.

A control strategy suitable for traction, in case of power train based on high speed permanent-magnet synchronous motor, is proposed in [18]. This control performs in a wide speed range, adding some additional blocks and the equations of the MTPV strategy, it mimics the same approach of the outer voltage loop solution. Nevertheless, even in [18], the issues related to torque transition from motoring to braking are not investigated. Finally, no tests on a real vehicle are shown [18]. Similar control strategies are presented in [19,20].

Improved MTPA and flux weakening control strategies that take into account magnetic saturation and the need of MTPV to obtain a full exploitation of the short term current rating can be found in [21].

Given the aforementioned scenarios, this paper presents two control schemes to properly address the torque transitions at high speed in FW operation: the former is based on the outer voltage loop [17] and the latter on [16]. Simulation and experiments of the proposed solutions are presented in the following paragraphs.

The paper is organized as in the following: Section 2 recalls the mathematical model at the basis of permanent-magnet synchronous motor control; Section 3 describes the flux weakening implementation approach and critical issues; Section 4 shows the traditional flux weakening control scheme and their experimental behaviour in case of abrupt torque transition at high speed; Section 5 shows the proposed optimized control strategies and their experimental results, Section 6 describes the experimental setup and the comparison of the different flux weakening approach in terms of power train performance.

## 2. PM Vector Control Equations

Equations (1) and (2) describe respectively the dynamic voltage model and the steady-state one in a rotor d-q reference frame:

$$\begin{cases} v_{sd} = R_s i_{sd} + L_{sd}\frac{di_{sd}}{dt} - \omega_e L_{sq} i_{sq} \\ v_{sq} = R_s i_{sq} + L_{sq}\frac{di_{sd}}{dt} + \omega_e \Lambda_{PM} + \omega_e L_{sd} i_{sd} \end{cases} \tag{1}$$

$$\begin{cases} v_{sd} = R_s i_{sd} - \omega_e L_{sq} i_{sq} \\ v_{sq} = R_s i_{sq} + \omega_e \Lambda_{PM} + \omega_e L_{sd} i_{sd} \end{cases} \tag{2}$$

where $i_{sd}, i_{sq}, v_{sd}, v_{sq}, \omega_e$ are the stator current and voltage components and the electrical angular speed, $\Lambda_{PM}$ is the PM flux, $R_s$ is the stator resistance, $L_{sd}$ and $L_{sq}$ are d- and q-axis inductances.

In the case of surface PM the rotor has no saliency and the $L_{sd}$ and $L_{sq}$ inductances are equal.

The terms $-\omega_e L_{sq} i_{sq}$ and $+\omega_e L_{sd} i_{sd}$ are the motional back EMF that can be compensated to decouple axes.

The torque is expressed in Equation (3) where the amplitude invariant form of Clarke's transformation was chosen:

$$T = \frac{3}{2}p\big[\Lambda_{PM}i_{sq} + (L_{sq} - L_{sd})i_{sd}i_{sq}\big] \tag{3}$$

where $p$ is the pole pair number.

In the constant torque region, where the motor operates in the MTPA mode, limits are due the current limit circle see Equation (4) only. On the contrary in the constant power region they rely on both current and voltage limits Equations (4) and (5). The voltage limit circle becomes an ellipse in the $i_{sd} - i_{sq}$ plane, due to the rotor saliency, see Equation (6) where $\omega_m$ is the rotor mechanical speed [22].

$$i_{sd}^2 + i_{sq}^2 \leq i_{Lim}^2 \tag{4}$$

$$v_{sd}^2 + v_{sq}^2 \leq v_{Lim}^2 \tag{5}$$

$$L_{sd}^2\left(i_{Lim} + \frac{\Lambda_{PM}}{L_{sd}}\right)^2 + L_{sq}^2 i_{sq}^2 \leq \left(\frac{V_{Lim}}{p\omega_m}\right)^2 \tag{6}$$

The maximum current depends on the inverter, while is the battery voltage and state of charge that limit the BUS voltage. It is to be noted that for a given machine, the voltage suitable for current regulators depends on machine mechanical speed $\omega_m$ as well since both the axes of the ellipse grow shorter as the mechanical speed increases. Equation (7) describes the intersection of the axis of the voltage ellipse.

$$i_{ch} = -\frac{\Lambda_{PM}}{L_{sd}} \tag{7}$$

The purpose of the motor control is to calculate the voltage at machine terminals that satisfies torque setpoint despite the mechanical speed.

## 3. Flux Weakening Implementation

From an ideal point of view, the driver sets the desired torque by means of the throttle. Thus, during acceleration, the torque and then the acceleration is proportional to the torque setpoint. This relationship is basically true in MTPA region, while in the FW one the torque decreases as the speed increases, even if the amplitude of current space vector is kept constant. An optimal control strategy should seek to approximate the aforementioned behaviour to fully exploit the FW motor capability.

FW in traction applications allows to extend the constant power region, eliminating the need for mechanical gear boxes, avoiding the volt-ampere over sizing of the power converter and increasing the power density of the system.

In order to implement FW strategy on an IPM motor, it is necessary to generate an appropriate negative $i_{sd}$ setpoint. In fact, a negative $i_{sd}$ current will produce a magnetic flux along the d-axis opposing the rated flux generated by PMs.

The strategy adopted to compute the $i_{sd}$ setpoint affects the motor performance and it must be designed considering: on one side the torque generation and on the other hand the voltage and current limitations. As known, an excessive FW action reduces the generated torque and then vehicle performance. On the other hand, a late start or a weak action may result in undesired torque drop according to the PI current regulator saturation.

The generation of the exact FW profile and consequently the $i_{sd}$ setpoint is a tough task since different variables (motor parameters, battery state of charge) and the motor operating conditions affect the optimal setpoint and the level of the FW action.

Strictly speaking, $i_{sd}$ and $i_{sq}$ could be evaluated once the desired torque and the flux magnitude are known. This methodology is know as FF and, as evident, it would be quite impossible to obtain optimal performance due to: motor parameters variations under different operation, battery state of charge value, magnetic saturation and ohmic voltage drop.

In [19], a robust FF FW control algorithm, based on mathematical motor model is proposed. Being $I_s$ the maximum inverter current and $V_{BUS}$ the actual BUS voltage, from Equations (4) and (6), it is possible to calculate the setpoints of $i_{sd-safe}$, $i_{sq-max}$ in function of the measured motor voltage and electric speed $\omega_e = p\omega_m$ as in Equation (8). The meaning of the chosen name $i_{sd-safe}$ will be clarify in the following.

$$\begin{cases} i_{sd-safe} = \dfrac{-\Lambda_{Pm}L_{sd} + \sqrt{\Lambda_{Pm}^2 L_{sd}^2 - (L_{sd}^2 - L_{sq}^2)(L_{sq}^2 I_{max}^2 + \Lambda_{PM}^2 - \frac{V_{BUS}^2}{\omega_m^2})}}{L_{sd}^2 - L_{sq}^2} \\ i_{sq-max} = \sqrt{I_s^2 - i_{sd}^2} \end{cases} \tag{8}$$

Figure 6 reports the results obtained by FF technique generation of $i_{sd}$ command. The results, in case of a FF calculated from rated motor parameters and measured battery pack voltage, are shown in red. While the results obtained in case of a 5% voltage underestimation are reported in blue. As expected, in FW region, torque and power values drop.

In both cases an open-loop FW strategy leads to results that are distant from those that could be got in an ideal situation, since the mechanical power is a long way off the constant trend.

In conclusion, some closed-loop corrections are necessary to fully exploit the power limit of IPM motors.

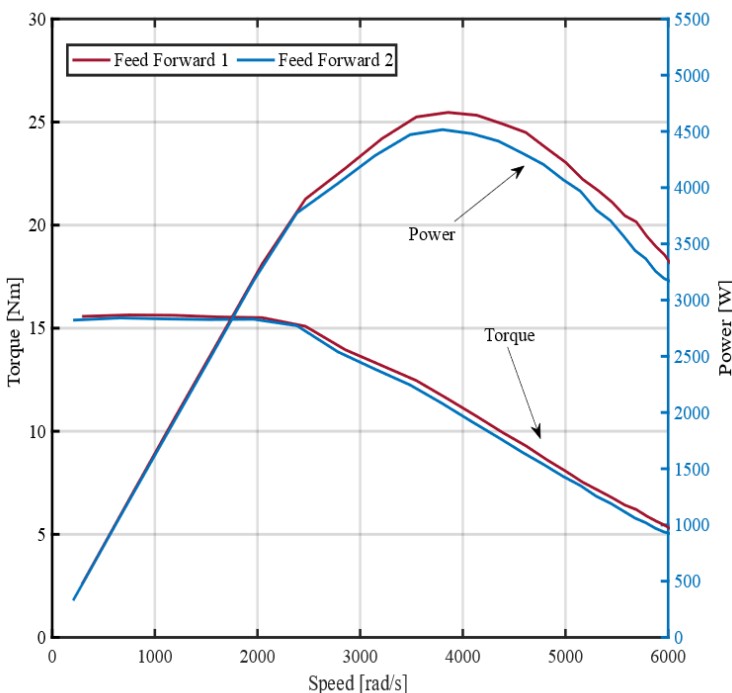

**Figure 6.** Feed forward results under two different battery voltage estimation. Red line actual battery voltage, blue line voltage underestimated of 5%.

## 4. Traditional Flux Weakening Control Scheme in Vehicle Applications

Many authors developed FW methods and control schemes based on the detection of the current error or of voltage saturation. The FW control schemes proposed in [15–17] have been investigated in this paper because of their simplicity and effectiveness.

Figure 4 shows the scheme proposed in [16] where MTPA strategy is carried out calculating the $i^*_{sd}, i^*_{sq}$ set points, following an equation or a LUT (to take into account magnetic saturation). The FW control is achieved with the $i_{sd}$ error ($\Delta i_{sd}$) between the d-axis current setpoint $i^*_{sd}$ and the actual value $\bar{i}_{sd}$. The $\Delta i_{sd}$ represents a valuable detection feedback. If d-axis current error $\Delta i_{sd}$ grows, the q-axis current setpoint $i^*_{sq}$ will be reduced, releasing voltage in order to compensate the error on $i_{sd}$. In this way, the current vector is kept inside the voltage limit ellipse and it is moved toward the $i_{sd}$ axis.

Figure 5 shows the FW control scheme proposed in [17]. In this paper, the setpoint of MTPA torque is transformed into the corresponding current setpoints $i_{a\_sp}$ in the d-q axis reference frame by function blocks. Then, $i_{a\_sp}$ setpoint is transformed from polar to rectangular coordinates: $i_{d\_sp}$ and $i_{q\_sp}$.

Both methods allow to take advantage of the flux weakening potential of the IPM electrical machine, at least in acceleration. In fact, both schemes rely on an integral action that drives the limits of the currents setpoints, thus some issues could appear in deceleration due to the integral dynamic of PI regulators. With the aim to test the FW performance of control schemes in [16,17], a complete model of the vehicle was developed in Matlab/Simulink environment. Both control schemes have been performed in acceleration and deceleration operations.

The simulation results of the Jahns' control scheme are shown in Figure 7, the MTPA trajectory has been evaluated keeping into account the saturation of the lamination following the $L_{sd}L_{sq}$ variation reported in Figure 8. The motor parameters used for the simulations are reported in Table 1. As expected, during fast torque transition the currents $i_{sd}$ and $i_{sq}$ are far from an ideal situation .

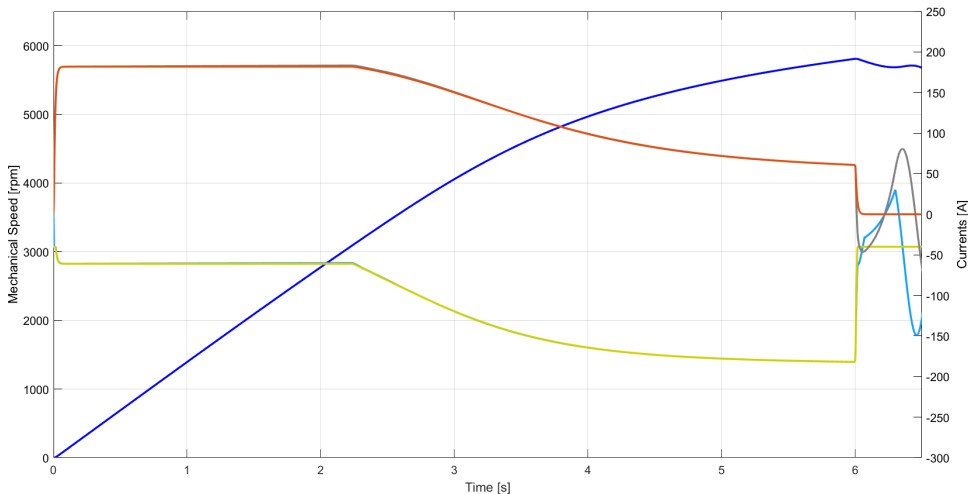

**Figure 7.** Simulation results with Jahns' traditional control scheme .

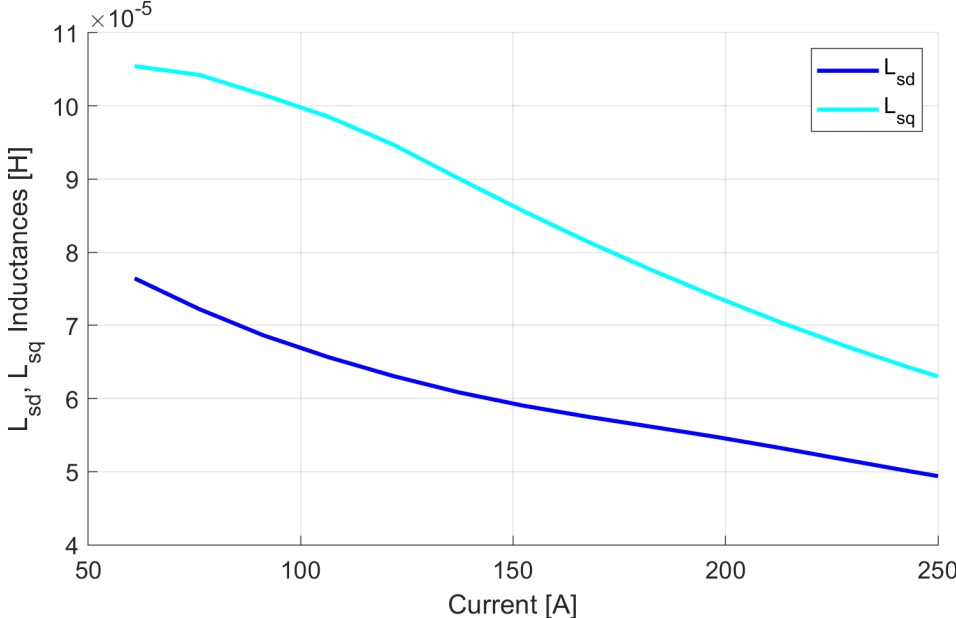

**Figure 8.** Motor inductances versus stator phase current.

**Table 1.** Main parameters for the Simulation.

| Description | Value | Unit of Measure |
|---|---|---|
| Motor External Diameter | 140 | mm |
| Motor Stack Length | 120 | mm |
| Motor Nominal Current | 160 | mm |
| Motor Nominal Torque | 15.8 | Nm |
| Motor base speed | 3000 | rpm |
| Motor Phase Resistance | 1.65 | mΩ |
| Motor Direct axis Inductance | 0.055 | mH |
| Motor Quadrature axis Inductance | 0.075 | mH |
| Motor Permanent magnet flux Linkage | 0.0128 | Wb |
| System Moment of inertia | 3 | kgm$^2$ |

Figures 9 and 10 show the experimental results in acceleration and deceleration at high speed of [16,17], respectively. As expected, both traditional FW control schemes perform quite good in acceleration, but show an unwanted behaviour when the torque setpoint goes to zero at high speed, as a results of the driver setpoint. This is the typical situation of a braking action.

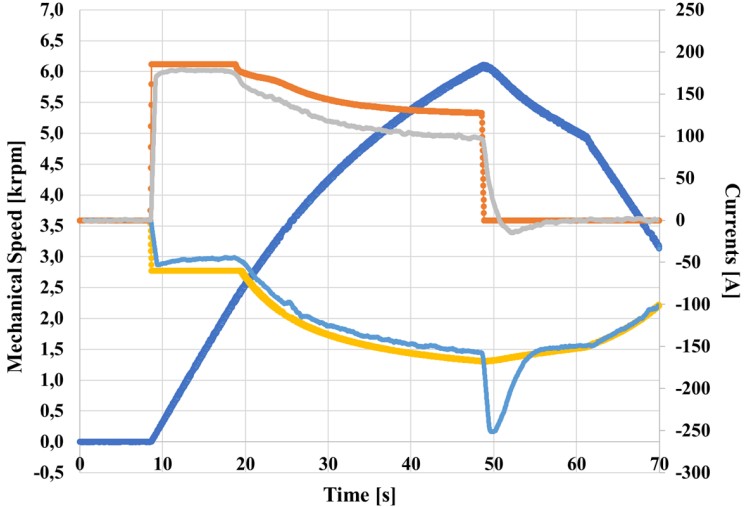

**Figure 9.** Experimental results of Jahns' control [16]. Angular mechanical speed profile (blue solid line) and $d-q$ currents signals: $i_{sd}$ setpoint (yellow solid line), motor $i_{sd}$ (light blue solid line), $i_{sq}$ setpoint (orange solid line) and motor $i_{sq}$ (grey solid line).

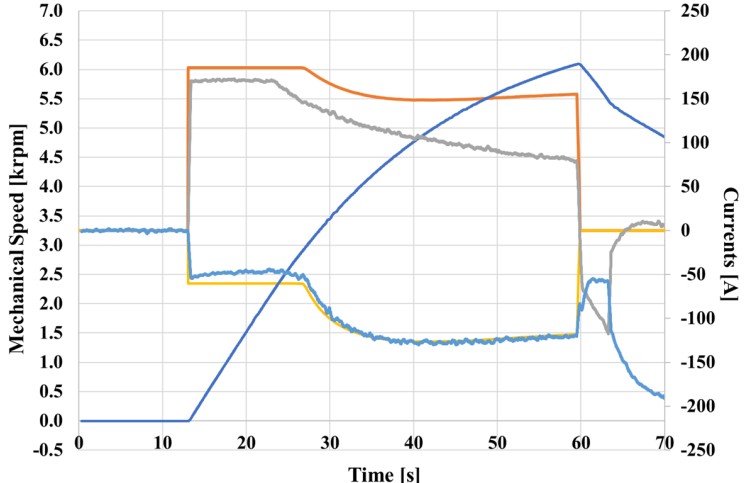

**Figure 10.** Experimental results of outer voltage control [17]. Angular mechanical speed profile (blue solid line) and $d-q$ currents signals: $i_{sd}$ setpoint (yellow solid line), motor $i_{sd}$ (light blue solid line), $i_{sq}$ setpoint (orange solid line) and motor $i_{sq}$ (grey solid line).

In Figures 11 and 12, it can be easily seen that the angle computed by the integrator (orange solid line) approaches the value computed for the MTPA, when torque setpoint goes instantly to zero. In [17], this behavior is due to the fact that the voltage error method changes sign and the integral action reduces it to its minimum angle value (about 18°) in braking condition. A similar phenomenon happens with FW control method of [16].

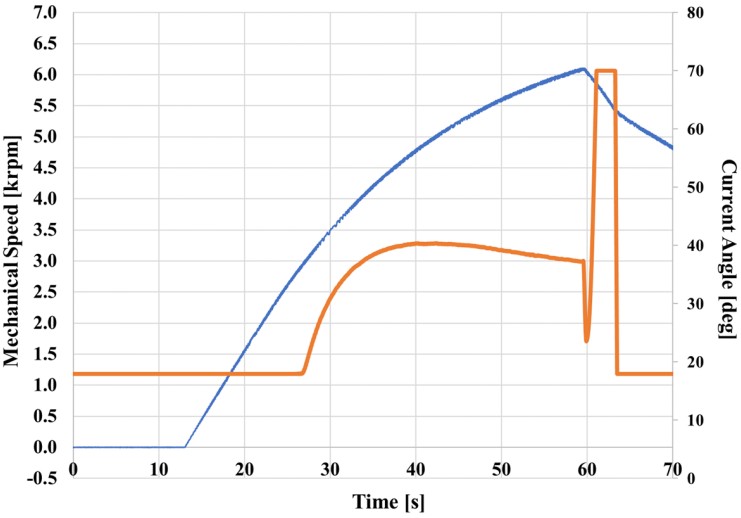

,

**Figure 11.** Experimental results of outer voltage control [17]. Current angle (orange solid line) and angular mechanical speed (blue solid line).

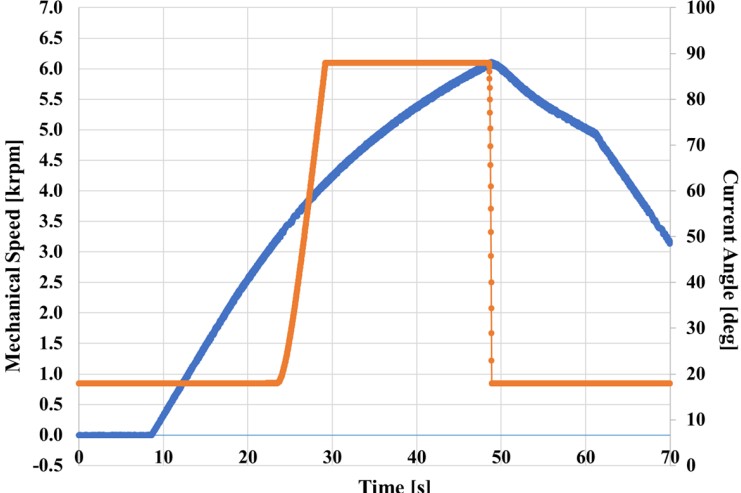

**Figure 12.** Experimental results of Jahns' control [16]. Current angle (orange solid line) and angular mechanical speed (blue solid line).

Since $i_{sd}$ setpoint depends solely on speed and voltage, a fast transition of the outer voltage error results in a wrong $i_{sd}^*$ reference value which is generally suitable for MTPA strategy. This behavior implies an error in current control and leads to a voltage error increase or an $i_{sd}$ error again.

Moreover, the current angle behavior of Figures 11 and 12 leads to an uncontrolled active braking torque, and a regenerating current will flow, uncontrolled, towards battery pack. Eventually , the braking feeling perceived by the driver will result uncomfortable as well: the early part of the brake mode would be out of control and it can be particularly harmful in some road conditions.

A different slope of the speed profile can be observed during braking phase with the outer voltage method, Figure 10. The experimental results with method of [16] are quite similar, see Figure 9.

## 5. Optimized Control Strategy for Flux Weakening Operation

This paper proposes a FF control loop based on the estimation of the $i_{sd}$ current setpoint, according to Equation (8). The objective of the proposed FF loop is improving

the robustness of control strategy during abrupt torque transition which could lead to unwanted brake operation mode of the electrical machine.

Experimental results show that the proposed approach leads to a good estimation of the d-axis current during braking operation in FW region. Especially, thanks to the FF control loop, the issue related to current angle calculation of Figures 11 and 12 is overcome.

The introduced FF loop works as follows: if an abrupt torque transition occurs in FW region and the $i_{sd}^*$ setpoint computed with the integral approach is lower than the $i_{sd-safe}$ value of Equation (8), then the $i_{sd-safe}$ is set as d-axis current setpoint. In so doing, the performance of the electrical machine as a motor is preserved and it won't move towards brake operation at high speed as a consequence of an unexpected torque transition. According to its action, the new $i_{sd}$ current contribution is named $i_{sd-safe}$. The corresponding improved control scheme proposed in [17] is shown in Figure 13.

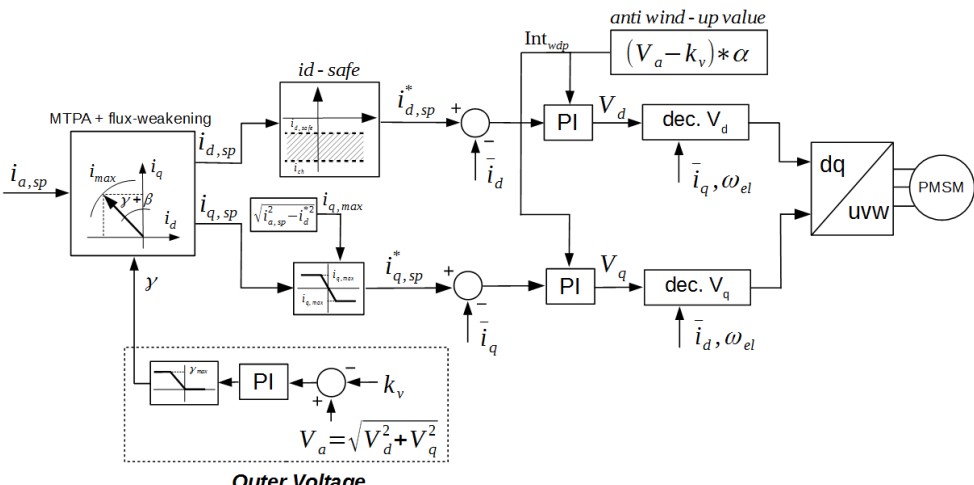

**Figure 13.** Control scheme of outer voltage method of [17] with the proposed $i_{sd-safe}$ strategy.

Figure 14 shows the experimental results of the improved outer voltage method of [17] with the $i_{sd-safe}$ FF contribution. Figure 15 shows the current angle in acceleration and deceleration from standstill to 6000 rpm and back. During deceleration, the value of the angle will be overridden by the $i_{sd-safe}$ of Equation (8) and the unwanted effect on the speed slope at the beginning phase of braking operation disappears providing a higher braking quality. .

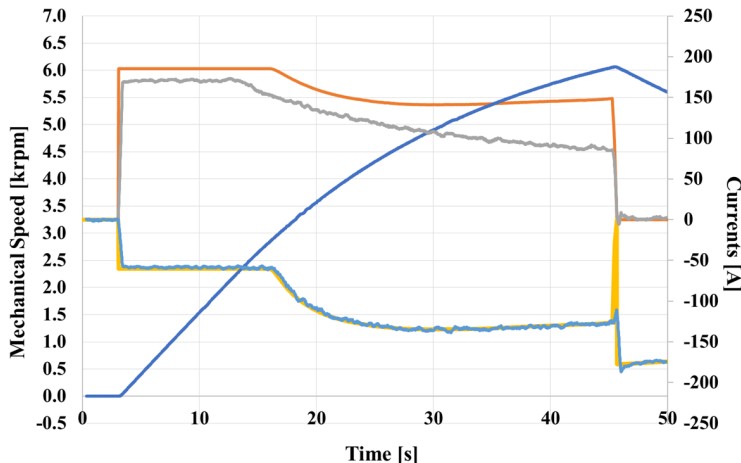

**Figure 14.** Experimental results of outer voltage method of [17] with proposed FW control. Speed profile and $i_{sd}, i_{sq}$ currents.

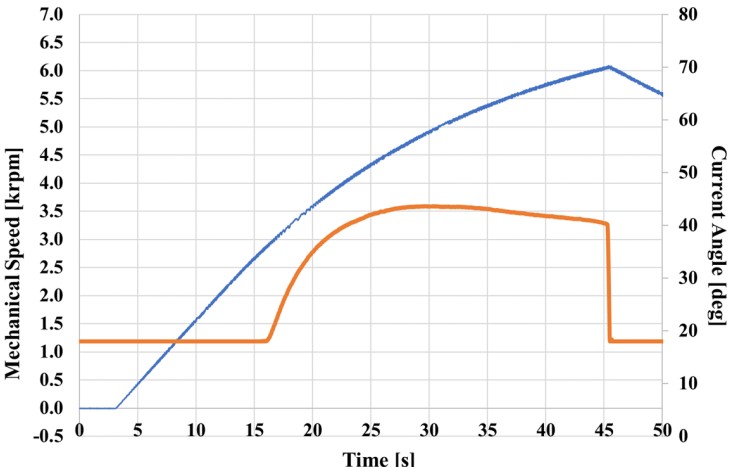

**Figure 15.** Experimental results of outer voltage method of [17] with proposed FW control: current angle (orange solid line) and angular mechanical speed (blue solid line) .

The presented FF solution was implemented also in [16] as well. In Figure 16 the additional $i_{sd-safe}$ block is shown, while Figure 17 shows the behaviour in the experimental setup in acceleration and deceleration from standstill to 6000 rpm and back. The shape of the current angle in [16] is very close to that shown in Figure 15 for the outer voltage scheme. The experimental results with improved Jahns' control scheme confirm the improved behavior and repetitiveness due to FF, $i_{sd-safe}$ loop in case of braking action.

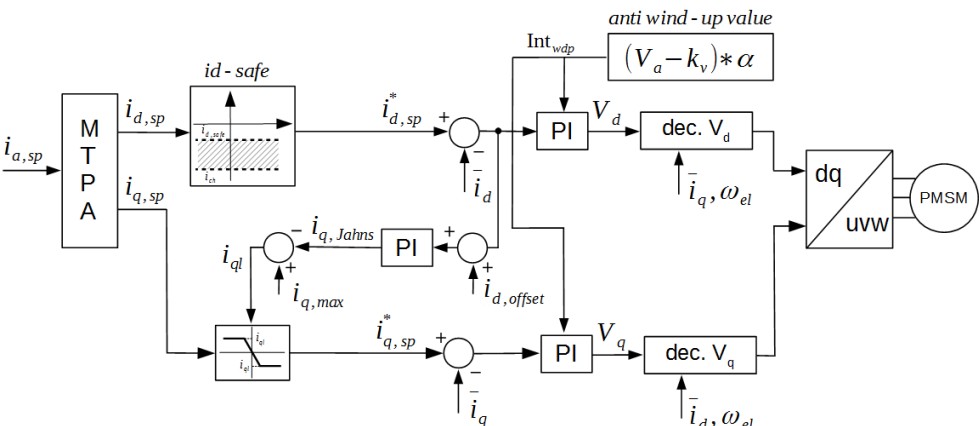

**Figure 16.** Proposed improved Jahns' control scheme.

In addition to the $i_{sd-safe}$ block, a particular voltage-based *anti-wind up* method has been introduced in both enhanced control schemes with the aim to further improve their performance in FW, with the aim to keep the desired voltage request around the the saturation edge.

The *anti-wind up* block multiplies by an appropriate gain, denoted as $\alpha$, the voltage error between the requested voltage vector amplitude and the available one. The output of the *anti-wind up* block is then used to limit the integral terms of current regulators. In this way, the deep saturation of current regulators, due to battery voltage limit, is avoided. Consequently the dynamic performance improves, since current regulators will ask from the system the voltage that it can be realized. This solution is not particularly suitable for outer voltage strategy, since, it derives the current angle from voltage saturation.

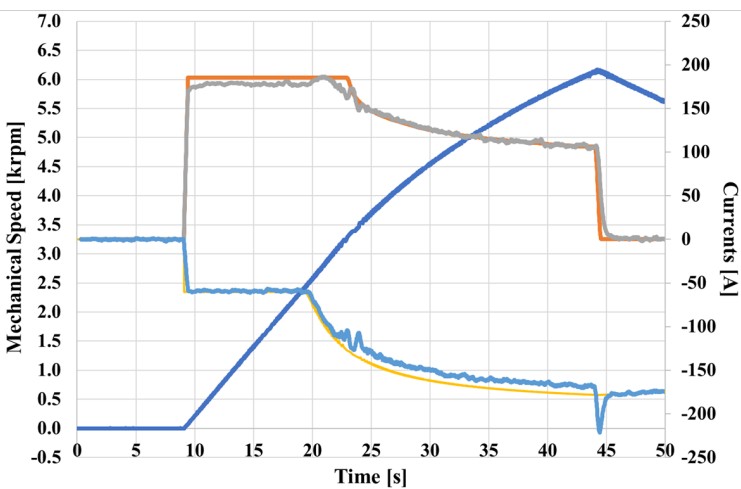

**Figure 17.** Experimental results of the improved Jahns' control scheme [16]: speed profile and $i_d$, $i_q$ currents signals.

## 6. Experimental Setup and Performance Comparison

A suitable test bed was implemented to test the proposed control strategies. The test bed was composed by a torquemeter, a power analyzer, the 12 slot 10 pole motor under test, a high current drive with programmable DSP, a battery pack and the active brake with a moment of inertia comparable to that of the actual vehicle. The complete setup is reported in Figure 18. The induction machine used as active brake was controlled by FOC control with appropriate position feedback with a speed close-loop. The motor under test was tolled with a current (torque) loop employing the proposed control schemes.

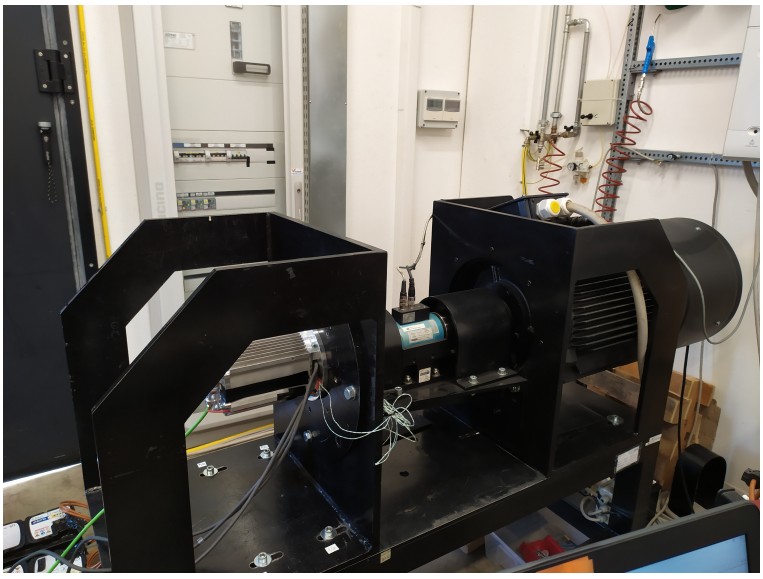

**Figure 18.** Test bench set up for experimental validation of the proposed control methods.

An ARM architecture DSP (STM32F407VET6) was employed to test the proposed control schemes implementing them in C code language. A set of hall-effect current transducers (LEM HC5F800-s) were used to measure the current in the control loops, a digital absolute magnetic encoder was employed to acquire the rotor position. The motor is cooled by natural air convection.

The control setpoints and measured quantities were acquired and logged by means of by STM-Studio software.

Figure 19 compares the experimental results of the feed-forward open loop approach. The enhanced outer voltage and the enhanced Jahns' torques and powers are reported

versus mechanical speed and it can be clearly noticed that the enhanced Jahns' control strategy outclasses either FF method, or outer voltage method.

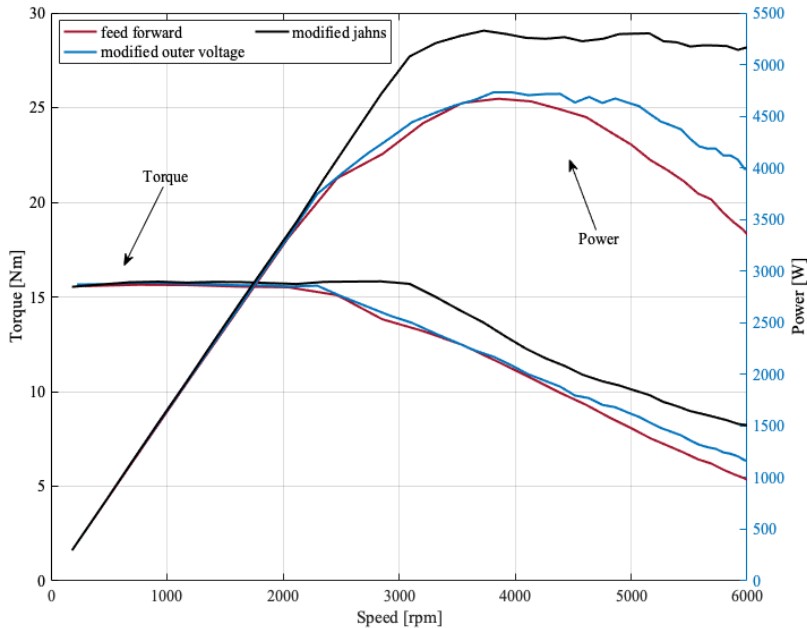

**Figure 19.** Comparison between the basic FF method and the proposed ones. FF (red line), enhanced outer voltage (blue line) and enhanced Jahns' method (black line). Experimental results.

## 7. Conclusions

This paper proposes different improved control strategies for flux weakening operation of IPM synchronous machines. The basis of the proposed FW schemes are some of the most traditional methods proposed in literature, and rely on the saturation detection of current PI regulators or on voltage saturation detection.

The performance of traditional FW control methods showed a good exploitation of the IPM motor when it accelerates, but because of the integral action of PI regulators used to calculate the current angle, some issues arise in deceleration operation if an abrupt torque transition occurs. This behavior has been proven both via simulations and experimental results.

Two additional feed forward contributions denoted $i_{sd-safe}$ and *anti-wind up* were shown in this paper. The FF loops were added to the traditional control schemes of [16,17] to overcome the aforementioned issues, improving their performance during deceleration operation at high speed.

The $i_{sd-safe}$ current acts on the $i_{sd}^*$ setpoint in accordance with motor angular speed and BUS voltage. While, the voltage-based *anti-wind up* keep away current regulators from saturation making full use of the battery voltage.

Both optimized methods have been validated at first by Matlab/Simulink simulations and then by extensive experimental tests on a dedicated test bench. The motor under test was a 12–10 slot-pole combination IPM synchronous machine for a light electric vehicle.

Experimental results showed that the FF actions improve the robustness and repetitiveness of the FW controls, in case of a sudden torque transition due to an abrupt braking action. In particular the $i_{sd-safe}$ FF loop sets a suitable current angle value, avoiding unwanted excessive braking that can occurs at high speeds improving the driver comfort and safety while the voltage-based *anti-wind up* allows to exploit constant Power region in a more efficient way.

**Author Contributions:** Conceptualization and methodology, G.F. and C.B.; experimental results G.F. and C.B.; validation, C.B. and G.F., investigation and data care, A.T.; writing—original draft preparation, C.B.; writing—review and editing and supervision, A.T. and C.B. All authors have read and agreed to the published version of the manuscript.

**Funding:** This research received no external funding.

**Data Availability Statement:** Data sharing not applicable.

**Conflicts of Interest:** The authors declare no conflict of interest.

## Abbreviations

The following abbreviations are used in this manuscript:

| | |
|---|---|
| IPM | Interior Permanent-Magnet |
| MTPA | Maximum Torque Per Ampere |
| FW | Flux Weakening |
| MTPV | Maximum Torque per Volt |
| PM | Permanent-Magnet |
| B-EMF | Back—ElectroMotive Force |
| FF | Feed Forward |
| LUT | Look-Up Tables |
| DPS | Digital Signal Processor |
| FOC | Field Oriented Control |

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
