# Peer review of "Improvement on Flux Weakening Control Strategy for Electric Vehicle Applications"

_applsci, doi:10.3390/app11052422_

Round 1
Reviewer 1 Report
Opinion in attached file

Author Response
Dear reviewer, we wish to thank you for the precious suggestions that helped us to make a better paper. In the following you’ll find all the improvements introduced on the basis of your observation.
Hoping to have met your indication we take the opportunity to convey our cordial greetings.
Review of "Improvement on Flux Weakening Control Strategy for Electric
Vehicle Applications".
Submitted for publication in Applied Sciences by: Claudio Bianchini , Giovanni
Franceschini, Ambra Torreggiani
Summary:
Existing and improved strategies for field weakening in a permanent magnet motor has been presented in this paper. Weakening is needed to operate the motor in the second part of the traction drive characteristics - constant power operation. Existing control strategies work well for motor acceleration and operation over a wide speed range. It causes significant current changes when the motor load torque changes abruptly - when switching from driving to braking the vehicle. Improvements involving the introduction of suitable coupling blocks into the control system has been proposed in this paper. Simulation results and residual experimental results confirming the accuracy of the introduced changes has been included in this paper.
Comments:
Authors should refer to the following comments before publication: 1. It is not specified in this paper for which equation parameters the
simulations were carried out.
As required a table has been added with all the main simulation parameters
- Traction drives are characterised by high inertia (the vehicle has high
inertia) - this was not taken into consideration in the experiment and
simulations, the experiment was carried out only on an inertia-free engine.
The inertia is included in the Matlab/Simulink model of the vehicle, the test bed is not inertia-free, the active brake and the transmission have an inertia comparable to that of the actual vehicles (a city scooter). As it can be seen from the experimental and simulation result, when the throttle is rapidly forced to zero the speed follows a slope that is due to the inertia of the system. Similar tests on the actual vehicle (a small city scooter) given the same results.
In every traction drive there are some slack in the transmission mechanisms,
this was not mentioned in the equations.
The adoption in the actual vehicle of helical tooth gears helps to reduce the Backlash beside, it is not possible to take into account all the imperfection of the real system and many of them are addressed by mechanical design. The test on the actual vehicle proved that this is a second order effect compared to inertia and/or other system dynamic effects.
Well known mathematical model assuming linearity of magnetic circuit
(constant inductances) has been presented in the paper. I think that the paper
would be much more significant if a field model of the motor was developed
and inductances were dependent on currents.
The mathematical model is based on the d-q axis model, anyway is quite easy to consider the effect of the magnetic saturation both in the Simulink model and in the experimental setup. As a matter of fact, the MTPA and the id safe contribution were evaluated mapping the isd-isq versus stator current. Some details and the Lsd Lsq profile Vs. current were added to the paper.

Reviewer 2 Report
This paper proposes an optimized control strategy for field weakening operation in EVs, specifically for IPMSM powertrains. As a starting point, this work considers classic field weakening solutions and introduces additional elements to improve the transient response under field weakening operation. Some simulation and experimental results validating the proposal are included within the manuscript.
This work deals with an interesting topic, but fails regarding the following critical points:
1) The literature review conducted within the paper regarding field weakening control strategies (and also the reviewed literature in general) is very limited and, in general, outdated. Thus, this work does not provide a sufficiently clear picture of recent advances in this field, and the novelty of the proposal cannot be justified that way. A number of solutions that could provide fast transient response in field weakening operation could be missing in the contextualization of this work.
It is mandatory to conduct an in-deep bibliographical review in order to provide a clear picture of the most recent advances in the field, highlighting the originality of the proposal.
2) The structure of the document should be greatly improved. On the one hand, writing must be better worked out. There are so many grammar mistakes. The usage of English should be reviewed by a proficient English user. The organization of ideas within paragraphs is also, many times, confusing. Introduction should be completely rewritten.
In many cases, important statements are provided within the paper without any citation that justifies it. This point is extremely relevant.
Some figures are not of the sufficient quality for publication in a JCR journal. For example, figures 1-3 are very simplistic. Figures providing the Matlab/Simulink model block implementation do not provide any information to the reader, are difficult to interpret, etc. Figure 20 does not provide any legend regarding x and y axes.
3) The following sentence is included in the introduction of the article (page 2 of the submitted manuscript):
“Many papers deal with the optimal magnetic flux quantity to obtain an optimal machine behavior, that can be substantially synthesized in a machine design that force the characteristic current on the edge of the current limit circle in the id iq plane. This design approach avoids the MTPV region to obtain the maximum exploitation of the power train.”
This point requires further clarification. From the industrial experience of the reviewer, at high speeds automotive IPMSMs generally operate in the Maximum Torque (or Flux) per Volt region. Thus, this design approach seems not to be common in the automotive industry. A summary of the papers dealing with such design approach should be cited in this work. Also, pros and cons of such design approach should be clearly stated.
All in all, it must be pointed out that this design approach is not common in real electric vehicle drives. Automotive IPMSMs operate in MTPV for a wide speed range. The presented solutions cannot provide MTPV operation.
4) Magnetic saturation is completely neglected in this work. In automotive surface mounted PMSMs this can be done due to the relatively large air gap, however, this is not true in automotive IPMSMs. In such applications considering magnetic saturation for both MTPA, field weakening and MTPV operation is mandatory. Not considering magnetic saturation leads to significant torque production errors and an overall efficiency reduction, among other issues.
5) In real EV applications, fast dynamic requirements during field weakening operation are not as stringent as it could be in other industrial applications. It must be pointed out that, in general, all torque references are highly ramped in real EVs to ensure driver’s and passengers’ comfort; thus, fast torque transient are not common. Thus and in the opinion of the reviewer, the proposal better suits for industrial drives with high dynamic requirements rather than for electric vehicle applications.
In the opinion of the reviewer, although the starting point of the work is interesting, it is recommended to be completely reworked and re-submitter. Also, considering the proposal, the reviewer suggest considering a general industrial drive scenario and not the EV one, as the proposal could not be useful in a real EV scenario (absence of MTPV operation, no magnetic saturation consideration, etc.).
Thus, I recommend to reject this article and I encourage authors to consider the previous points to prepare a completely new version of the manuscript for submission, focusing on electric drive applications in general.
Author Response
Dear reviewer, we wish to thank you for the precious suggestions that helped us to make a better paper. In the following you’ll find all the improvements introduced on the basis of your observation.
Hoping to have met your indication we take the opportunity to convey our cordial greetings.
This paper proposes an optimized control strategy for field weakening operation in EVs, specifically for IPMSM powertrains. As a starting point, this work considers classic field weakening solutions and introduces additional elements to improve the transient response under field weakening operation. Some simulation and experimental results validating the proposal are included within the manuscript.
This work deals with an interesting topic, but fails regarding the following critical points:
1) The literature review conducted within the paper regarding field weakening control strategies (and also the reviewed literature in general) is very limited and, in general, outdated. Thus, this work does not provide a sufficiently clear picture of recent advances in this field, and the novelty of the proposal cannot be justified that way. A number of solutions that could provide fast transient response in field weakening operation could be missing in the contextualization of this work.
It is mandatory to conduct an in-deep bibliographical review in order to provide a clear picture of the most recent advances in the field, highlighting the originality of the proposal.
Some references were added to the state of the art including recent works to prove the novelty of the proposed paper. Meanwhile the paper was checked and some parts rewritten to comply with reviewer notes 1) and 2).
2) The structure of the document should be greatly improved. On the one hand, writing must be better worked out. There are so many grammar mistakes. The usage of English should be reviewed by a proficient English user. The organization of ideas within paragraphs is also, many times, confusing. Introduction should be completely rewritten.
In many cases, important statements are provided within the paper without any citation that justifies it. This point is extremely relevant.
Some references were added in the paper to support the statements.
Some figures are not of the sufficient quality for publication in a JCR journal. For example, figures 1-3 are very simplistic. Figures providing the Matlab/Simulink model block implementation do not provide any information to the reader, are difficult to interpret, etc. Figure 20 does not provide any legend regarding x and y axes.
Figure 1,2,3 and 4 have been improved with more details, some mistakes have been fixed. The captions of the figures have been updated to specify if results are from experimental tests or simulations. Figure 10 and figure 11 have been added to improve the quality of the paper. Fig. 20 has been adjusted.
3) The following sentence is included in the introduction of the article (page 2 of the submitted manuscript):
“Many papers deal with the optimal magnetic flux quantity to obtain an optimal machine behavior, that can be substantially synthesized in a machine design that force the characteristic current on the edge of the current limit circle in the id iq plane. This design approach avoids the MTPV region to obtain the maximum exploitation of the power train.”
This point requires further clarification. From the industrial experience of the reviewer, at high speeds automotive IPMSMs generally operate in the Maximum Torque (or Flux) per Volt region. Thus, this design approach seems not to be common in the automotive industry. A summary of the papers dealing with such design approach should be cited in this work. Also, pros and cons of such design approach should be clearly stated.
Many papers deal with the optimal design line, one of the most cited is:
- L. Soong and T. J. E. Miller, "Theoretical limitations to the field-weakening performance of the five classes of brushless synchronous AC motor drive," 1993 Sixth International Conference on Electrical Machines and Drives (Conf. Publ. No. 376), Oxford, UK, 1993, pp. 127-132.
The key point of the optimal design line is to avoid the MTPV because MTPV implies the reduction of the maximum current amplitude to follow the trajectory toward the maximum speed. When the optimal design line is followed during the machine design the maximum current amplitude can be exploited till the maximum speed (theoretical infinite speed). Some references were added to the paper to better clarify this point.
All in all, it must be pointed out that this design approach is not common in real electric vehicle drives. Automotive IPMSMs operate in MTPV for a wide speed range. The presented solutions cannot provide MTPV operation.
4) Magnetic saturation is completely neglected in this work. In automotive surface mounted PMSMs this can be done due to the relatively large air gap, however, this is not true in automotive IPMSMs. In such applications considering magnetic saturation for both MTPA, field weakening and MTPV operation is mandatory. Not considering magnetic saturation leads to significant torque production errors and an overall efficiency reduction, among other issues.
The magnetic saturation is not neglected, the MTPA trajectory was evaluated considering the variation of the inductances with the stator currents. The flux weaking method adopted and proposed in the paper not requires on an accurate magnetic model since they are able to control the available voltage or the error on the id current loop.
5) In real EV applications, fast dynamic requirements during field weakening operation are not as stringent as it could be in other industrial applications. It must be pointed out that, in general, all torque references are highly ramped in real EVs to ensure driver’s and passengers’ comfort; thus, fast torque transient are not common. Thus and in the opinion of the reviewer, the proposal better suits for industrial drives with high dynamic requirements rather than for electric vehicle applications.
An emergency braking at high speed implies to rapidly move the torque setpoint to zero. In deep flux weakening region this implies to reduce the amplitude of the q axis current to zero keeping the id current at a suitable value (different from zero) to counteract the Back-EMF controlling unwanted braking torque from the electric motor. You are right: the situation is different from industrial application, but also in this case high dynamic or if you prefer high quality performance are required. We clarify this aspect in the paper and, instead of “high dynamic”, that could confuse the reader, we described in more appropriate way the torque transition (motor to braking) that could generate problems to vehicle dynamics at high speed (deep FW conditions).
In the opinion of the reviewer, although the starting point of the work is interesting, it is recommended to be completely reworked and re-submitter. Also, considering the proposal, the reviewer suggest considering a general industrial drive scenario and not the EV one, as the proposal could not be useful in a real EV scenario (absence of MTPV operation, no magnetic saturation consideration, etc.).
Thus, I recommend to reject this article and I encourage authors to consider the previous points to prepare a completely new version of the manuscript for submission, focusing on electric drive applications in general.

Reviewer 3 Report
Dear Authors,
the proposed manuscript presents and compares flux weakening strategies for permanent magnet synchronous machines. To improve the quality of the paper, I have some comments:
- According to the abstract, an optimized flux weakening strategy is proposed in the paper. Unfortunately, it is hard to understand for me, why optimal and what is the objective function of the proposed method. Please provide more detailed description to clarify this.
- "Internal" is more rarely used expression than "interior" as a type of permanent magnet synchronous motor. It would be better to use expression "interior".
- According to the title of figure 1., it presents the constant torque and power regions. However, only torque curves can be seen. Power curves would be also useful for the reader. Moreover, the labels are not consistent in fig. 1. Only the dimension is given in the speed label.
- In page 3, paragraph 3, it is mentioned that "few papers and studies dealt with a deep investigation of the fast torque of the fast torque setpoint transition at high speed". Please provide some references for this statement.
- Please clarify the novelty of the proposed manuscript over the previous papers in Section 1.
- Please provide consistent notations in the paper. For example, different notations are used for the voltage components in equations 1, 2 and in the text. The same mistake can be seen in the symbols of currents. Moreover, the symbol of rotor electrical speed is not introduced after eq. 2.
- It is mentioned in the abstract, that Matlab simulations and experimental tests are also applied to investigate the performances of the flux weakening strategies. The experimental setup is presented in Section 6, however, experimental results are presented before (such as figures 10, 11, 15, 18). Please clarify which results come from the simulations and which from the experiments.
- The axis labels are missing in fig. 20. Please check this.
- To compare the field weakening strategies in section 6., efficiency maps would be welcome.
Kind regards
Author Response
Dear reviewer, we wish to thank you for the precious suggestions that helped us to make a better paper. In the following you’ll find all the improvements introduced on the basis of your observation.
Hoping to have met your indication we take the opportunity to convey our cordial greetings.
Dear Authors,
the proposed manuscript presents and compares flux weakening strategies for permanent magnet synchronous machines. To improve the quality of the paper, I have some comments:
- According to the abstract, an optimized flux weakening strategy is proposed in the paper. Unfortunately, it is hard to understand for me, why optimal and what is the objective function of the proposed method. Please provide more detailed description to clarify this.
We modify the abstract and the introduction to better clarify this point, we also add some additional references
- "Internal" is more rarely used expression than "interior" as a type of permanent magnet synchronous motor. It would be better to use expression "interior".
We agree: “internal” was substituted with “interior” as suggested.
- According to the title of figure 1., it presents the constant torque and power regions. However, only torque curves can be seen. Power curves would be also useful for the reader. Moreover, the labels are not consistent in fig. 1. Only the dimension is given in the speed label.
The Figure 1 has been updated following the remark
- In page 3, paragraph 3, it is mentioned that "few papers and studies dealt with a deep investigation of the fast torque of the fast torque setpoint transition at high speed". Please provide some references for this statement.
Some references were added
- Please clarify the novelty of the proposed manuscript over the previous papers in Section 1.
- Please provide consistent notations in the paper. For example, different notations are used for the voltage components in equations 1, 2 and in the text. The same mistake can be seen in the symbols of currents. Moreover, the symbol of rotor electrical speed is not introduced after eq. 2.
The equations have been corrected to provide consistent notations
- It is mentioned in the abstract, that Matlab simulations and experimental tests are also applied to investigate the performances of the flux weakening strategies. The experimental setup is presented in Section 6, however, experimental results are presented before (such as figures 10, 11, 15, 18). Please clarify which results come from the simulations and which from the experiments.
Figure 10 has been added as comparison between experimental and simulation results, all the figure captions have been updated to specify if they show simulation or experimental results.
- The axis labels are missing in fig. 20. Please check this.
The figure has been modified following your suggestion
- To compare the field weakening strategies in section 6., efficiency maps would be welcome.
Very good remark, evaluating and comparing the efficiency is not a trivial task because there are multiple isd, isq combination for a given working point of the mechanical characteristic, this aspect is out of the purpose of the paper, it could be the topic of a future work.
Kind regards

Round 2
Reviewer 2 Report
The reviewed version of the manuscript is of much better quality than the first submitted version, as the problem is now better contextualized.
However, the following points should bedressed for its publication:
1) Matlab/Simulink block diagrams (figures 8 and 9) should be omitted, as they do not provide any information for the readers. They are extremelly confusing and do not add anything relevant to the paper.
2) Figures 2 and 3 could be merged into a single figure.
3) From the technical point of view, altough the paper clarifies that it is possible to design a machine without MTPV, this is clearly not the current trend in the automotive industry. Thus, I suggest to incorporate a sentence like this in the justification of page 2: "Thus, machines which avoid MTPV operation could be also considered for electric vehicle applications (...)" and then focus the research on such machines.
4) The title of section 4 should omit the "vehicle applications", as the presented field weakening solutions are of general purpose or classic (this last is properly stated at the conclusions of the manuscript).
5) The bibliography should be further updated (this is a weak point of the manuscript). One posibility is to review more modern field weakening control approaches that are needed to operate also in MTPV mode. Then, it could be justified that for the machine design that is considered in this paper classic or traditional approaches are more appealing due to their simplicity. Such broarder view of the problem is missing in this work.
6) The technical part of the manuscript is appropriate and does not require further modifications.
Author Response
Dear Reviewer,
Thank you for your remarks, In the following you will find all the improvements introduced based on your observations.
Hoping to have met your indications we take the opportunity to convey our cordial greetings.
The reviewed version of the manuscript is of much better quality than the first submitted version, as the problem is now better contextualized.
However, the following points should bedressed for its publication:
1) Matlab/Simulink block diagrams (figures 8 and 9) should be omitted, as they do not provide any information for the readers. They are extremelly confusing and do not add anything relevant to the paper.
We removed the figures as requested
2) Figures 2 and 3 could be merged into a single figure.
We merged figure 2 and 3 as requested
3) From the technical point of view, altough the paper clarifies that it is possible to design a machine without MTPV, this is clearly not the current trend in the automotive industry. Thus, I suggest to incorporate a sentence like this in the justification of page 2: "Thus, machines which avoid MTPV operation could be also considered for electric vehicle applications (...)" and then focus the research on such machines.
We have incorporated the suggested sentence and extend the state-of-the-art analysis to better clarify this point
4) The title of section 4 should omit the "vehicle applications", as the presented field weakening solutions are of general purpose or classic (this last is properly stated at the conclusions of the manuscript).
The title of section 4 was changed as requested
5) The bibliography should be further updated (this is a weak point of the manuscript). One posibility is to review more modern field weakening control approaches that are needed to operate also in MTPV mode. Then, it could be justified that for the machine design that is considered in this paper classic or traditional approaches are more appealing due to their simplicity. Such broarder view of the problem is missing in this work.
The biography was further extended with recent works taking into account also the MTPV strategy and we better clarified the link between the machine design and the need of the MTPV strategy.
6) The technical part of the manuscript is appropriate and does not require further modifications.

Round 3
Reviewer 2 Report
After both reviews, the article has the quality to be published in the Applied Sciences Journal.